# Medical Admission Prediction Score (MAPS); a simple tool to predict medical admissions in the emergency department

Muhammad Zahid[1,2,3☯], Adeel Ahmad Khan [4☯]*, Fateen Ata [4☯], Zohaib Yousaf[5], Vamanjore Aboobacker Naushad[1], Nishan K. Purayil[1], Prem Chandra[6], Rajvir Singh[6], Anand Bhaskaran Kartha[1,2,3], Abdelnaser Y. Awad Elzouki[1,2,3], Dabia Hamad S. H. Al Mohanadi[1,2,3], Ahmed Ali A. A. Al-Mohammed[1,2,3]

**1** Department of Medicine, Hamad General Hospital, Hamad Medical Corporation, Doha, Qatar, **2** College of Medicine, Qatar University, Qatar, Qatar, **3** Weill Cornell Medicine, Ar-Rayyan, Qatar, **4** Department of Endocrinology, Hamad General Hospital, Hamad Medical Corporation, Doha, Qatar, **5** Department of Medicine, Reading Hospital-Tower Health, West Reading, PA, United States of America, **6** Department of Medical Research, Medical Research Center, Academic Health System, Hamad Medical Corporation, Doha, Qatar

☯ These authors contributed equally to this work.
* adeel_1026@yahoo.com, akhan40@hamad.qa

## Abstract

### Introduction

Overcrowding in the emergency departments (ED) is linked to adverse clinical outcomes, a negative impact on patient safety, patient satisfaction, and physician efficiency. We aimed to design a medical admission prediction scoring system based on readily available clinical data during ED presentation.

### Methods

In this retrospective cross-sectional study, data on ED presentations and medical admissions were extracted from the Emergency and Internal Medicine departments of a tertiary care facility in Qatar. Primary outcome was medical admission.

### Results

Of 320299 ED presentations, 218772 were males (68.3%). A total of 11847 (3.7%) medical admissions occurred. Most patients were Asians (53.7%), followed by Arabs (38.7%). Patients who got admitted were older than those who did not (p <0.001). Admitted patients were predominantly males (56.8%), had a higher number of comorbid conditions and a higher frequency of recent discharge (within the last 30 days) (p <0.001). Age > 60 years, female gender, discharge within the last 30 days, and worse vital signs at presentations were independently associated with higher odds of admission (p<0.001). These factors generated the scoring system with a cut-off of >17, area under the curve (AUC) 0.831 (95% CI 0.827–0.836), and a predictive accuracy of 83.3% (95% CI 83.2–83.4). The model had a sensitivity of 69.1% (95% CI 68.2–69.9), specificity was 83.9% (95% CI 83.7–84.0), positive

**Data Availability Statement:** Data sharing requires permission from the Ministry of Public Health, Qatar. Any request for data requisition can be made to the Medical Research Center (MRC) at Hamad

Medical Corporation Qatar, which will seek legal permission from the Ministry of Public Health before sharing the data. The MRC can be contacted via the following: Email: research@hmc.org.qa Phone: +974- 439 2440/6162/6163/6164.

**Funding:** Article publication charges were provided by Qatar National Library. The funders had no role in study design, data collection and analysis, decision to publish, or preparation of the manuscript.

**Competing interests:** The authors have declared that no competing interests exist.

predictive value (PPV) 14.2% (95% CI 13.8–14.4), negative predictive value (NPV) 98.6% (95% CI 98.5–98.7) and positive likelihood ratio (LR$^+$) 4.28% (95% CI 4.27–4.28).

## Conclusion

Medical admission prediction scoring system can be reliably applied to the regional population to predict medical admissions and may have better generalizability to other parts of the world owing to the diverse patient population in Qatar.

## Introduction

Timely and adequate care in the emergency department (ED) is a critical determinant of hospital quality of care and efficiency. EDs around the world are becoming busier and more congested as a result of a variety of factors such as increased patient volume, access blockage, lack of adequate primary healthcare services, increased patient complexity and acuity, lack of awareness of proper use of emergency services, extended stay of border patients in ED, and budget constraints, among others [1–3]. Overcrowding in the ED is linked to increased inpatient mortality and hospital length of stay, patient dissatisfaction due to long wait times, and a negative impact on physician productivity and morale [3, 4]. ED visits in the United Kingdom (UK) have increased by 40% in the last 15 years, with 20% of attendees eventually admitted to hospital and another 10% referred to outpatient services [5, 6]. According to a report published by the United States Department of Health and Human Services, over 143 million ED visits occurred in 2018 in the United States with more than 20 million resulting in admissions [7].

Clinical decision-making concerning medical admission in the ED is time-consuming and requires a wide range of skills and expertise, particularly when dealing with patients of varying acuity who require multiple simultaneous decisions in a highly demanding and charged environment. The admission prediction model is one solution to predict admission or discharge at the front gate on presentation. This prediction, coupled with early notification of the admitting ward, may decrease wait time in ED and subsequently relieve overcrowding [8–10]. Several models, including the Glasgow admission prediction score (GAPS), Sydney triage to admission risk tool (START) and START2 model, A novel prediction model by Clare Parker, have been developed using both logistic regression and machine learning with moderate to high accuracy [11–15]. However, no model has been widely adopted as simplicity and accuracy have a steep tradeoff [11]. Furthermore, ethnic and geographic differences have a significant impact on disease onset, clinical course, and outcomes. Some of the most common comorbidities, such as diabetes mellitus (DM), hypertension (HTN), and chronic kidney disease (CKD) have an earlier age of onset and a higher prevalence in Asian and Middle Eastern populations than in Western populations [16–18]. Similarly, many other critical factors considered in generating admission prediction scores vary greatly depending on ethnicity and region. This greatly limits the utility of the developed scores to the specific population and hence cannot be generalized.

Hamad Medical Corporation (HMC) is Qatar's principal provider of secondary and tertiary healthcare services. ED at HMC is one of the busiest in the world, with 1200 to 1500 attendees a day, facing similar challenges as other ED departments in the world [19]. The goal of this study was to create a scoring system for predicting the likelihood of hospitalization based on clinical data readily available at the time of arrival to the ED for patients presenting with medical complaints. This tool will help with efficiency and improve the quality of clinical decision-

making for ED patients who are likely to be admitted to the medicine department, ultimately improving patient flow.

## Materials and methods

### Study design and setting

We used a retrospective, cross-sectional design to achieve study aims. Data were retrieved from the electronic medical records (EMR) with the help of the patient's health card number (HC Number) from the Health Information Department (HIM). The data retrieved included demographic characteristics, comorbid conditions, vital signs at presentation, information regarding a recent discharge from the hospital, and mortality during the current encounter.

### Study population

Pre-specified data of all adult patients who attended the ED from 1st January 2019 to 31st December 2019, and fulfilled the study inclusion criteria, were collected.

### Inclusion criteria

Patients $\geq$ 14 years of age who presented to the ED with medical complaints and were evaluated with vital signs measurement during the study period were included.

### Exclusion criteria

Patients who left the ED before a decision of admission could be made, those presenting with non-medical complaints (trauma, fractures, gynaecological, acute surgical, psychiatric, ophthalmologic, and otorhinolaryngologic complaints), those who did not have vital signs recorded and who died in the ED before a decision to admit could be made, were excluded.

### Patient involvement

The study did not involve any patients directly as the data was obtained from electronic medical records (EMR) Cerner®. The authors did not have access to information that could identify individual participants during or after data collection.

### Primary outcome

The study's primary outcome was to develop a scoring system to predict admission to the medical ward.

### Statistical analysis

We used descriptive statistics to summarize and determine the sample cohort's characteristics and distribution of participants' data. The normally distributed data were reported as mean with standard deviation (SD), whereas median and inter-quartile range (IQR) were used for skewed or non-normal data distribution. We summarized the categorical data using frequencies and proportions. We assessed the associations between two or more categorical data variables using the Pearson Chi-Square ($\chi^2$) test. We analyzed quantitative data between the two independent groups (medical admission vs non-admission) using unpaired t test. Non-parametric Mann Whitney U test was applied when the data distribution is skewed.

Univariate and multivariate logistic regression methods were used to assess the predictive values of each predictor or risk factor (such as age, Gender, RR, PR, SPO2, GCS, Systolic BP, number of co-morbidities and discharge within 30 days) for dichotomous outcome variable

medical admission (yes/no). For multivariate regression models, predictors were considered if found statistically significant (at the P<0.10 level) in univariate analysis or if determined to be clinically important. The results of logistic regression analyses were reported as odds ratio (OR) with 95% confidence intervals (CIs). The number of risk factors identified in multivariable logistic regression analyses was summarized to compute weighted risk scores and generate a clinically applicable decision-making rule for predicting medical admission upon the first encounter with the patient in the ED. The regression coefficients were divided by the smallest coefficient and then rounded to the nearest integer to derive a simple-to-compute risk score. A two-sided P value <0.05 was considered to be statistically significant.

The primary data analysis in our research study was directed to determine the predictive accuracy of the various predictors and covariates in predicting hospital admission using weighted risk score classification. For this, the sensitivity, specificity, positive and negative predictive values of these parameters were calculated, using admission to medical floors as the point of reference. A receiver operating characteristic (ROC) curve was calculated using significant predictors (as determined via multivariate regression) to derive the best suitable cut-off values and to assess model discrimination and predictive accuracy. Because sensitivity and specificity were considered equally important, the best cutoff points were determined using Youden's index which maximizes sensitivity and specificity. ROC curves provide a comprehensive and visually attractive way to summarize the accuracy of predictions. The ROC curve shows the tradeoff between sensitivity and specificity. It is a widely adopted method to detect the performance of a developed test, which classifies subjects into two categories: medical admission and no medical admission.

Bootstrapping (re-sampling) method, introduced by Efron (1979), was used for calculating bias-corrected percentile intervals (BCa) using 100 re-samples to make a traditional multivariate regression model a realistic model for the population [20].

All statistical analyses were done using statistical packages SPSS version 27.0 (Armonk, NY: IBM Corp) and Epi-info (Centers for Disease Control and Prevention, Atlanta, GA) software. The statistical analysis for this study was performed by the same statisticians who performed the statistical analysis for another research study in which a scoring system for foreign body aspiration in children was evaluated and formulated [21]. The same statistical analysis strategies was adopted in developing the potential scoring system in predicting hospital admission in this study as well.

### Ethics declaration

This work is original, has not been, and is not under consideration for publication in any other Journal. All authors have reviewed and approved the final version of the manuscript. The study was conducted in full compliance with the principles of the "Declaration of Helsinki," and Good Clinical Practice (GCP). The study was approved by the Medical Research Centre (MRC), Qatar (MRC-01-20-1094).

### Participant consent

Informed consent was waived by the IRB, as this study was a retrospective data review of medical records.

## Results

### Baseline demographics

A total of 320,299 of patients visiting the ED fulfilling the study criteria were included for the analysis (Fig 1), of which 11,847 (3.7%) were admitted to the medical ward. Men made up 68%

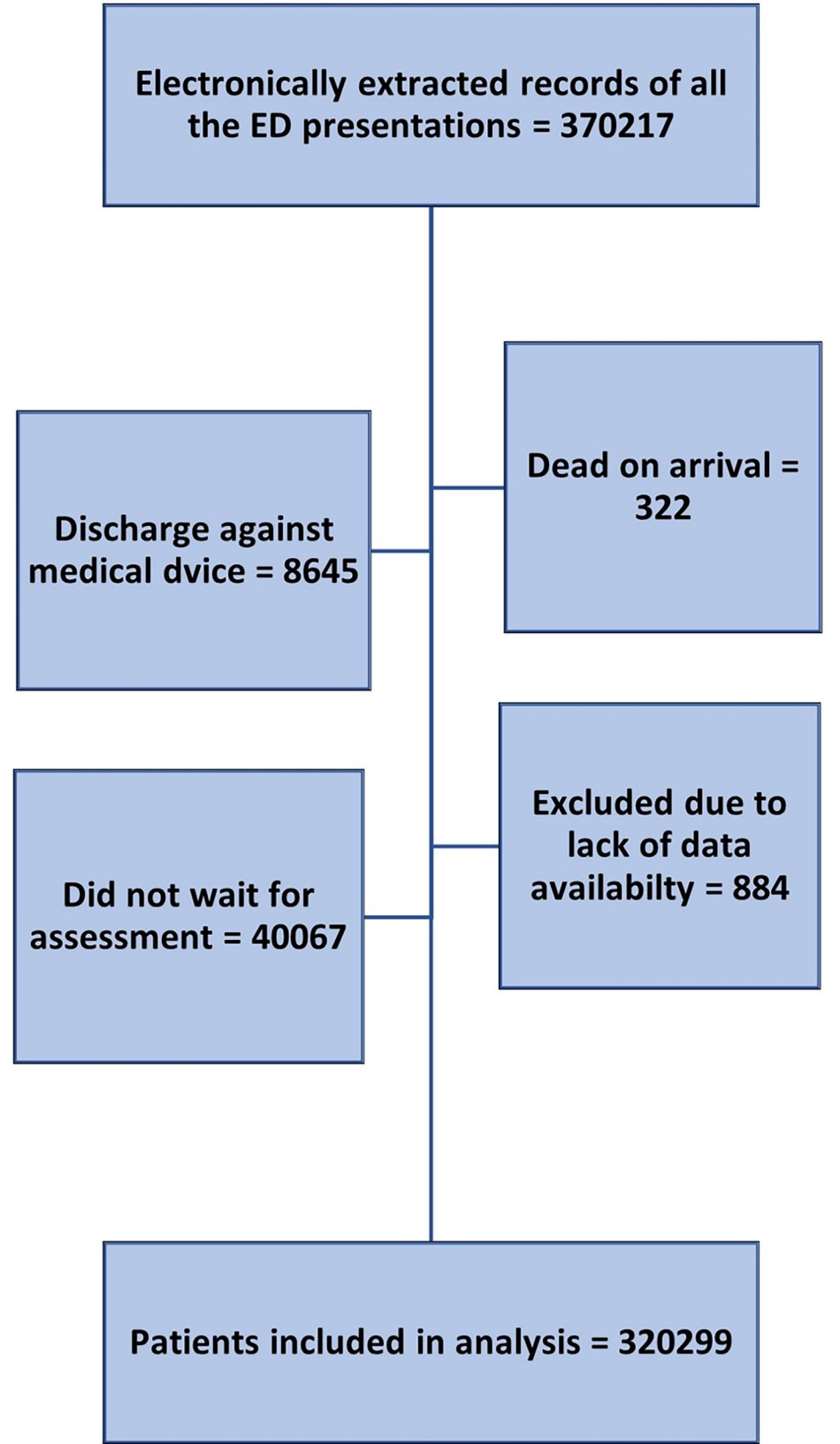

**Fig 1. Flow diagram of the patient inclusion process.**

(218,772) of the total cohort and 56.8% (6733) of the medical admission group. 172198 (53.7%) patients were Asians, 124062 (38.7%) were Arabs, 18405 (5.74%) were Africans, and 2331 (0.7%) were Europeans. The patients in the admitted group were older when compared to the overall group (49.7 ± 19.1 vs 35.1 ± 14.1years). The mean number of comorbid conditions was higher among the admission group (2.1 ± 1.8 vs. 0.5 ± 1) (P<0.001). The most common comorbidities among the admission group were hypertension (HTN) (52.09%), diabetes mellitus (DM) (47%), asthma (23.2%), chronic kidney disease (CKD) (22.1%), cerebrovascular accidents (CVA) (17.1%), ischemic heart disease (IHD) (10.9%) and malignancy (10.3%). More patients in the admitted group had a recent discharge (within the last 30 days) as compared to those in the not admitted group (21% vs. 3.7%) (p <0.001) **Table 1**.

### Baseline hemodynamic parameters

There was no difference in the mean temperature among the admission and the non-admission groups (37.2 ± 9.1˚C and 37.1 ± 11.2˚C, respectively) (P 0.2). The mean respiratory rates were 19.8 ± 5.8 breaths per minute (PM) vs. 18.2 ± 6.9 breaths PM in the admission and the non-admission groups (P<0.001). The mean pulse rate in the admission group was 87.3 ± 20.3 beats PM compared to 84.3 ± 26.2 beats PM in the non-admission group (P<0.001). The mean oxygen saturation (97.8 ± 3.4% in the admission group vs. 99 ± 1% in the non-admission group) and mean systolic blood pressure (132.6 ± 28.5 mmHg in the admission group vs. 128.2 ± 18 mmHg in the non-admission group) also differed significantly (P<0.001) **Table 1**.

### Risk factors for medical admission

The results of the univariate analysis are summarized in **Table 2**. Age > 60 years (OR 8.3), female gender (OR 1.67), an increasing number of comorbidities (OR ranging from 4.5 in patients with up to 2 comorbidities to an OR of 25.4 in patients with more than five comorbidities), discharge within the last 30 days (OR 6.9), and worse vital signs at presentations were all associated with the higher risk of admission to the medical ward (p<0.001). All these factors retained statistical significance when analyzed in the multivariate logistic regression model (p<0.001). Finally, the multivariate logistic regression results were used to generate the scoring system for predicting medical ward admission **Table 3**. The cut-off value for predicting admission to the medical ward using ROC was >17, and the area under the curve (AUC) was 0.831 (95% CI 0.827–0.836) with a predictive accuracy of 83.3% (95% CI 83.2–83.4) (**Fig 2**). The sensitivity of the model at this cut-off was 69.1% (95% CI 68.2–69.9), specificity was 83.9% (95% CI 83.7–84.0), positive predictive value (PPV) 14.2% (95% CI 13.8–14.4), negative predictive value (NPV) 98.6% (95% CI 98.5–98.7) and positive likelihood ratio (LR+) 4.28% (95% CI 4.27–4.28). (S1 Table).

Based on results from 100 boot-strap samples, boot-strap confidence intervals were relatively narrower than the traditional model. The corrected model had less than 1% bias in all the parameters of the traditional model coefficients **Table 4**.

### Discussion

In this study, we developed a scoring system predicting hospital admission for patients presenting to ED using nine readily available variables upon the first encounter with the patient. These included two demographic variables (age and sex) and seven clinical variables (pulse rate, respiratory rate, systolic blood pressure, oxygen saturation, Glasgow coma scale, number of comorbidities, and hospitalization in the last 30 days). At a cut-off score of >17, our model has a very good predictive performance and discriminative ability with an AUC of 0.831 (95% CI 0.827, 0.836) [22]. Admission prediction can be readily made via this scoring system during

**Table 1. Baseline characteristics of all ED presentations and comparison of admitted vs. non-admitted patients (N = 320299).** Data are presented as Mean ± SD and numbers with percentages as appropriate.

| Baseline characteristics | Study Cohort | Admitted | Not Admitted | P-value* |
|---|---|---|---|---|
| **Total Number** | 320299 | 11847 (3.7) | 308452 (96.3) | |
| **Age** | 35.15 ± 14.1 | 49.74 ± 19.1 | 34.58 ±13.5 | <0.001 |
| ≤60 years | 301339 (94.1) | 8247 (69.6) | 293092 (95.02) | <0.001 |
| >60 years | 18929 (5.9) | 3600 (30.38) | 15329 (4.97) | |
| **Gender** | | | | |
| Male | 218772 (68.3) | 6733 (56.8) | 212039 (68.7) | <0.001 |
| Female | 101524 (31.7) | 5114 (43.1) | 96410 (31.2) | |
| **Vitals** | | | | |
| Temperature (˚C) | 37.1 ± 11.1 | 37.2 ± 9.1 | 37.1±11.2 | 0.200 |
| **RR (breaths pm)** | 18.26 ± 6.92 | 19.8 ± 5.8 | 18.2 ± 6.9 | <0.001 |
| 12–24 | 309678 (99.1) | 10939 (3.5) | 298739 (96.5) | <0.0001 |
| 10–11 or 24–34 | 2395 (0.8) | 733 (30.6) | 1662 (69.4) | |
| ≤9 or ≥35 | 520 (0.2) | 147 (28.3) | 373 (71.7) | |
| **PR (beats pm)** | 84.06 ± 23.11 | 87.3 ± 20.3 | 84.3 ± 26.2 | <0.001 |
| 51–100 | 279527 (88.4) | 8375 (3) | 271152 (97) | <0.001 |
| 101–110 | 20564 (6.5) | 1440 (7) | 19124 (93) | |
| 111–130 or 41–50 | 13767 (4.4) | 1559 (11.3) | 12208 (88.7) | |
| >130 or ≤40 | 2475 (0.8) | 440 (17.8) | 2035 (82.2) | |
| **SPO$^2$ (%)** | 98.96 ± 1.23 | 97.86 ± 3.43 | 99.01 ±1.03 | <0.001 |
| >94 | 313703 (99.3) | 10883 (3.5) | 302820 (96.5) | <0.001 |
| 88–94 | 1975 (0.6) | 722 (36.6) | 1253 (63.4) | |
| <88 | 316 (0.1) | 238 (75.3) | 78 (24.7) | |
| **GCS** | 14.96 ± 0.58 | 14.50 ± 1.86 | 14.98 ±0.44 | <0.001 |
| 15 | 296909 (99.1) | 10562 (3.6) | 286347 (96.4) | <0.001 |
| 13–14 | 1170 (0.4) | 463 (39.6) | 707 (60.4) | |
| 10–12 | 775 (0.3) | 415 (53.5) | 360 (46.5) | |
| ≤9 | 721 90.2) | 398 (55.2) | 323 (44.8) | |
| **Systolic BP (mmHg)** | 128.39 ± 18.52 | 132.6 ± 28.5 | 128.2 ±18 | <0.001 |
| 101–160 | 301839 (95.6) | 10187 (3.4) | 291652 (96.6) | <0.001 |
| 161–180 or 81–100 | 9760 (3.1) | 944 (9.7) | 8816 (90.3) | |
| 181–220 or 71–80 | 3751 (1.2) | 593 (15.8) | 3158 (84.2) | |
| >220 or ≤70 | 366 (0.1) | 120 (32.8) | 246 (67.2) | |
| **Co-morbidities** | | | | |
| IHD | 7351 (2.3) | 1303 (10.9) | 6048 (1.9) | <0.001 |
| CVA/TIA | 4813 (1.5) | 2028 (17.1) | 2785 (0.9) | <0.001 |
| Hypertension | 45547 (14.2) | 6172 (52.09) | 39375 (12.7) | <0.001 |
| Diabetes | 39049 (12.2) | 5579 (47) | 33470 (10.85) | <0.001 |
| CKD | 9540 (3) | 2623 (22.1) | 6917 (2.2) | <0.001 |
| COPD | 6927 (2.2) | 851 (7.1) | 6076 (1.96) | <0.001 |
| Malignancy | 4301 (1.3) | 1228 (10.3) | 3073 (0.99) | <0.001 |
| Asthma | 40972 (12.8) | 2757 (23.2) | 38215 (12.38) | <0.001 |
| Chronic lung disease (unspecified) | 25765 (8) | 2030 (17.13) | 23735 (7.69) | <0.001 |
| **Number of comorbidities (Mean ± SD)** | 0.58 ± 1.14 | 2.07 ± 1.86 | 0.52 ± 1.06 | <0.001† |
| **Discharge within the last 30 days** | 14002 (4.4) | 2499 (21.1) | 11503 (3.7) | <0.001 |
| **Number of co-morbidities** | | | | |

(*Continued*)

**Table 1.** (Continued)

| Baseline characteristics | Study Cohort | Admitted | Not Admitted | P-value* |
|---|---|---|---|---|
| 0 | 231577 (72.3) | 3316 (27.99) | 228261 (74) | <0.001 |
| 1–2 | 64736 (20.2) | 3956 (33.39) | 60780 (19.7) | |
| 3–4 | 18836 (5.9) | 3188 (26.9) | 15648 (5.07) | |
| ≥5 | 5150 (1.6) | 1387 (11.7) | 3763 (1.21) | |

RR: Respiratory Rate; PR: Pulse Rate; SPO2: Oxygen saturation; GCS: Glasgow Comma Scale; BP: Blood pressure; CKD: Chronic Kidney Disease; COPD: Chronic Obstructive Pulmonary Disease; IQR: Interquartile Range; Quantitative data were expressed as mean ± standard deviation.

*P-values computed using Pearson Chi-Square and unpaired t tests.

† Non-parametric Mann Whitney U test.

triage in the ED. It does not require sophisticated computer programs or additional information that would not be available until sometime after the presentation.

ED triage scales, including the Canadian triage acuity scale, the Manchester triage scale, the Australasian triage scale, and the emergency severity index, are complex scoring methods with limited use by less experienced ED clinicians and in resource-constrained settings [23–26]. The Modified Early Warning Score (MEWS) is another simple tool that can be obtained from assessment of vital signs at presentation [27]. Studies have shown a linear relationship between ED MEWS and admission to the hospital [27–32]. In addition to using vital signs similar to MEWS, we have incorporated additional variables (age, gender and number of nomorbid condition) that are usually readily available at the time of ED presentation to generate a more comprehensive Medical Admission Prediction Score (MAPS). n prediction score (MAPS).

Cameron et al. developed an admission prediction model (Glasgow Admission Prediction Model, GAPS) using age, vital signs, triage category, referral by general physician (GP), arrival in ambulance and admission in the last year [33]. Although the authors used admission in the last year as one of the factors predicting a current admission, they showed a linear relationship of admission within the next 28 days for patients discharged from the ED [33]. Multiple studies have reported discharge within 30 days as a strong predictor of readmission. Various 30-day readmission rates are reported, ranging from 7% in the UK to 20% in the USA [34–36]. Factors that predict 30-day readmission might differ from shorter or longer durations between admissions [37]. 30-day readmission is also the most studied time duration and best reflects resource utilization compared to shorter or longer durations between admissions [37]. For the reasons mentioned above, it is also the most commonly used readmission duration indicator by various healthcare and health insurance policymakers [34, 38]. As the authors of this study agree with better predictability of discharge within the last 30 days rather than longer durations with readmissions and healthcare burden, it was also used in the current model.

Dinh et al. developed a Sydney triage-to-admission risk tool (START) to predict hospital admission from the ED in Australia using age, ambulance arrival, triage category, admission within 30 days, ED arrival time, and the presenting problem as variables [12]. In the START prediction tool, the history of admission in the last 30 days obtained a weightage of +3 risk score based on an odds ratio of 1.93 (95%CI 1.90, 1.96, $p < 0.001$) [12]. In our model, this variable has an OR of 2.68 (95% CI 2.53, 2.84, $p < 0.001$) and carries a weightage of +7 risk score. Both GAPS and START models used regional triage system limiting their broader applicability.

Parker et al. created a predicted model using electronic clinical records of all ED visits to a large urban hospital in Singapore over ten years. Eleven variables including four demographics (age, gender, ethnicity, and proximity of patients' home to study site), four administrative

**Table 2. Univariate logistic regression analysis of factors predicting admission to the medical ward.**

| Baseline characteristics | Unadjusted Odds ratio (95% CI) | P-value* |
|---|---|---|
| **Age** | | |
| ≤60 years | 1.0 (reference) | <0.001 |
| >60 years | 8.3 (7.96–8.67) | |
| **Gender** | | |
| Male | 1.0 (reference) | <0.001 |
| Female | 1.67 (1.61–1.73) | |
| **Temperature (˚C)** | (0.99–1.002) | <0.001 |
| **RR (breaths pm)** | | |
| 12–24 | 1.0 (reference) | |
| 10–11 or 24–34 | 12.04 (11.01–13.1) | <0.001 |
| <9 or >35 | 10.8 (8.9–13.03) | <0.001 |
| **PR (beats pm)** | | |
| 50–100 | 1.0 (reference) | |
| 101–110 | 2.4 (2.3–2.58) | <0.001 |
| 111–130 or 41–50 | 4.1 (3.9–4.4) | <0.001 |
| >130 or <40 | 7 (6.3–7.8) | <0.001 |
| **SPO2 (%)** | | |
| >94 | 1.0 (reference) | |
| 88–94 | 16 (14.6–17.6) | <0.001 |
| <88 | 84.9 (65.7–109.7) | <0.001 |
| **GCS** | | |
| 15 | 1.0 (reference) | |
| 13–14 | 17.7 (15.7–20) | <0.001 |
| 10–12 | 31.2 (27.1–36) | <0.001 |
| <9 | 33.4 (28.8–38.7) | <0.001 |
| **Systolic BP (mmHg)** | | |
| 100–160 | 1.0 (reference) | |
| 160–180 or 81–100 | 3 (2.85–3.28) | <0.001 |
| 180–220 or 71–80 | 5.4 (4.9–5.9) | <0.001 |
| >220 or <70 | 14 (11.1–17.4) | <0.001 |
| **Number of co-morbidities** | | |
| 0 | 1.0 (reference) | |
| 1–2 | 4.5 (4.3–4.7) | <0.001 |
| 3–4 | 14 (13.3–14.8) | <0.001 |
| ≥ 5 | 25.4 (23.6–27.2) | <0.001 |
| **Discharge within the last 30 days (yes)** | 6.9 (6.6–7.2) | <0.001 |

RR: Respiratory Rate; PR: Pulse Rate; SPO2: Oxygen saturation; GCS: Glasgow Comma Scale; BP: Blood pressure; CI: Confidence Interval.

*Univariate logistic regression method.

variables (day of the week, shift time of presentation, mode of arrival, and whether or not visit occurred during public school holidays) and three clinical variables (fever, triage category, and the number of visits to ED in previous calendar year) were selected for data analysis. The study used too many variables to develop a model, making it less generalizable and applicable [14]. To make our admission prediction model more practical and time-efficient, some factors deemed less significant to the local population were omitted. This resulted in a model different than the previously used prediction models elsewhere, as discussed above. For instance, we did

**Table 3. Multivariate logistic regression model for determining significant predictors and deriving weighted risk scores for hospital admission.**

| Predictor | Admitted Cases N (%) | Regression Coefficient | Adjusted Odds ratio† (95%CI) | P-Value* | Weight Risk score‡ |
|---|---|---|---|---|---|
| **Age (years)** | | | | | |
| ≤60 years | 2.7% | Reference | 1.0 (reference) | | |
| >60 years | 19.0% | 0.629 | 1.88 (1.77, 2.01) | <0.001 | 5 |
| **Gender** | | | | | |
| Male | 3.1% | Reference | 1.0 (reference) | <0.001 | |
| Female | 5.0% | 0.133 | 1.14 (1.10, 1.19) | | 1 |
| **RR (breaths pm)** | | | | | |
| 12–24 | 3.5% | Reference | 1.0 (reference) | | |
| 10–11 or 24–34 | 30.6% | 1.27 | 3.55 (3.12, 4.03) | <0.001 | 10 |
| <9 or 35–49 | 28.3% | 1.37 | 3.95 (2.89, 5.39) | <0.001 | 10 |
| **PR (beats pm)** | | | | | |
| 50–100 | 3% | Reference | 1.0 (reference) | | |
| 101–110 | 7% | 0.79 | 2.19 (2.06, 2.35) | <0.001 | 6 |
| 111–130 or 41–50 | 11.3% | 1.29 | 3.65 (3.41, 3.90) | <0.001 | 10 |
| >130 or <40 | 17.8% | 1.82 | 6.14 (5.39, 7.01) | <0.001 | 14 |
| **SPO2 (%)** | | | | | |
| >94 | 3.5% | Reference | 1.0 (reference) | | |
| 88–94 | 36.6% | 1.05 | 2.86 (2.54, 3.22) | <0.001 | 8 |
| <88 | 75.3% | 2.60 | 13.52 (9.68, 18.87) | <0.001 | 20 |
| **GCS** | | | | | |
| 15 | 3.6% | Reference | 1.0 (reference) | | |
| 13–14 | 39.6% | 2.05 | 7.78 (6.72, 9.02) | <0.001 | 15 |
| 10–12 | 53.5% | 1.98 | 7.26 (6.09, 8.66) | <0.001 | 15 |
| <9 | 55.2% | 2.56 | 12.92 (10.63, 15.7) | <0.001 | 19 |
| **Systolic BP (mmHg)** | | | | | |
| 100–160 | 3.4% | Reference | 1.0 (reference) | | |
| 160–180 or 81–100 | 9.7% | 0.27 | 1.31 (1.21, 1.42) | <0.001 | 2 |
| 180–220 or 71–80 | 15.8% | 0.67 | 1.96 (1.77, 2.17) | <0.001 | 5 |
| >220 or <70 | 32.8% | 1.68 | 5.39 (4.17, 6.95) | <0.001 | 13 |
| **Number of Co-morbidities** | | | | | |
| 0 | 1.4% | Reference | 1.0 (reference) | | |
| 1–2 | 6.1% | 1.18 | 3.25 (3.09, 3.42) | <0.001 | 9 |
| 3–4 | 16.9% | 1.91 | 6.78 (6.37, 7.22) | <0.001 | 14 |
| > = 5 | 26.9% | 2.09 | 8.04 (7.34, 8.81) | <0.001 | 16 |
| **Discharge within 30 days** | | | | | |
| No | 3.0% | Reference | 1.0 (reference) | | |
| Yes | 17.8% | 0.99 | 2.68 (2.53, 2.84) | <0.001 | 7 |

RR: Respiratory Rate; PR: Pulse Rate; SPO2: Oxygen saturation; GCS: Glasgow Comma Scale; BP: Blood pressure; SD: Standard Deviation

* Multivariate logistic regression method; OR: odds ratio; CI: confidence interval.

†Adjusted for all other potential covariates found in the univariate logistic regression analysis.

‡The scores were computed and derived by dividing the regression coefficients of the included predictors by the smallest regression coefficient and then rounding them to the nearest integer. For each patient, a sum score was calculated by adding the scores that correspond to the predictors and characteristics of the patient.

not include arrival by ambulance and ED arrival time to create a more translatable model applicable to our population. Qatar has a very diverse population with many young single male laborers. Only 10% of the 423,389 patients who received treatment at the emergency department at HGH in 2012 arrived via emergency medical services. A vast majority of

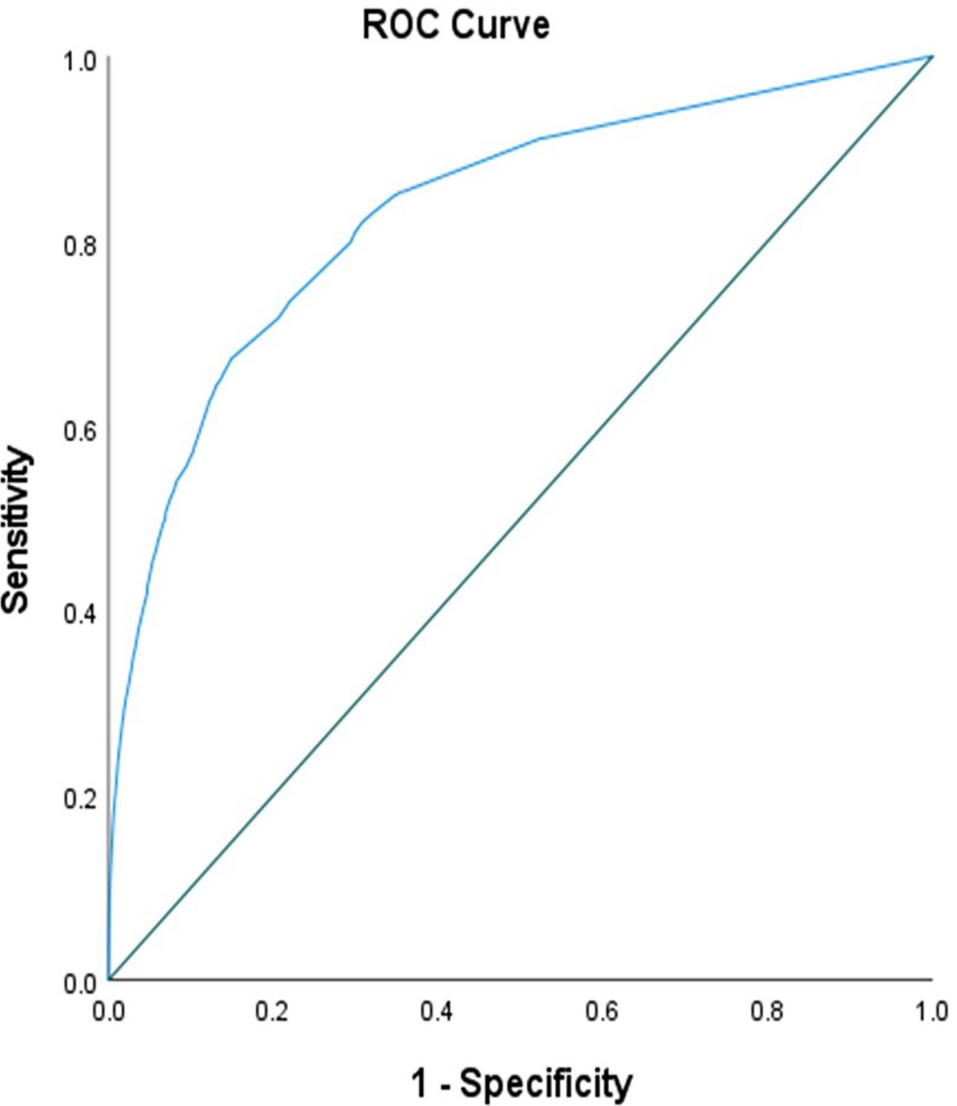

**Fig 2. Receiver operating characteristic (ROC) curve of the Medical Admission Prediction Score.**

patients arrive at the ED's front door at HGH by foot or private transportation (EMS), thus reducing the correlation of the mode of transportation to ED with the need for admission [39].

Age is an essential factor in predicting hospital admission and mortality[10, 33, 40, 41]. In the START prediction model, OR increased from 3.31 (95% CI 3.25, 3.37) for 60–79 years old to 6.01 (95% CI 5.89, 6.13) for > 80 years old patients [12]. Lucke et al. compared the performance of the admission prediction model for patients younger and older than 70 years. They found that the analytical capability of the models was greater for patients under 70. Still, the model for older patients was better at finding the group of patients very likely to be admitted [42]. In our model, patients > 60 years old had an OR of 1.88 (95% CI 1.77, 2.01, p< 0.001) for hospital admission compared to those ≤60 years old.

## Strengths and limitations of the study

The greatest strength of this study lies in its large sample size and diverse population. With a large Asian and Arabic population in the study, the results can be reliably applied to many

**Table 4. Bootstrap (100 resampling) multivariate logistic regression analysis to rule out bias in estimates.**

| Variable | Bias | SE Corrected | Bootstrap 95% C.I. | |
|---|---|---|---|---|
| **Gender (male)** | 0.003 | 0.023 | 1.079 | 1.196 |
| **Age (>60 years)** | 0.004 | 0.032 | 1.756 | 2.020 |
| **Number of Comorbidities** | | | | |
| 1–2 | -0.003 | 0.027 | 3.022 | 3.397 |
| 3–4 | 0.0 | 0.036 | 6.203 | 7.367 |
| > = 5 | -0.003 | 0.05 | 7.156 | 8.785 |
| **PR (beats pm)** | | | | |
| 101–110 | 0.0 | 0.034 | 2.071 | 2.340 |
| 111–130 or 41–50 | 0.001 | 0.038 | 3.404 | 3.975 |
| >130 or <40 | 0.002 | 0.062 | 5.371 | 6.973 |
| **Systolic BP (mmHg)** | | | | |
| 160–180 or 81–100 | 0.002 | 0.036 | 1.228 | 1.426 |
| 180–220 or 71–80 | -0.002 | 0.051 | 1.742 | 2.160 |
| >220 or <70 | -0.002 | 0.143 | 4.055 | 7.411 |
| **RR (breaths pm)** | | | | |
| 10–11 or 24–34 | -0.007 | 0.079 | 2.930 | 4.104 |
| <9 or 35–49 | 0.007 | 0.135 | 3.065 | 5.452 |
| **SPO2 (%)** | | | | |
| 88–94 | -0.003 | 0.066 | 2.573 | 3.340 |
| <88 | 0.002 | 0.195 | 9.507 | 21.370 |
| **GCS** | | | | |
| 13–14 | -0.003 | 0.078 | 6.646 | 8.962 |
| 10–12 | 0.003 | 0.097 | 5.977 | 8.611 |
| <9 | -0.018 | 0.088 | 10.350 | 15.135 |
| **Discharge within 30 days (yes)** | 0.005 | 0.028 | 2.547 | 2.852 |

RR: Respiratory Rate; PR: Pulse Rate; SPO2: Oxygen saturation; GCS: Glasgow Comma Scale; BP: Blood pressure; CKD: Chronic Kidney Disease; COPD: Chronic Obstructive Pulmonary Disease; SD: Standard Deviation; CI: Confidence interval

countries with a similar patient population where such prediction tools have not been validated. Many Asian and Arab countries face healthcare resource limitations amid a significant patient load in the EDs. Our easy-to-use prediction model, which requires no additional resources, will significantly improve patient care in the EDs where it is most needed. The inclusion of patients belonging to 176 different nationalities adds significantly to the applicability of our scoring system in other parts of the world as well. The boot-strapping method further validated the scoring system generated based on this study, improving its reliability. One important limitation of the study is a lack of prospective validation of the scoring system on real-world patients.

## Conclusion

A scoring system based on patient demographics, initial vital signs, comorbidities, and information regarding discharge within the last 30 days can help in predicting hospital admission in patients >14 years of age presenting to ED with medical complaints and can help to reduce the time in clinical decision-making process. Further prospective studies are required to incorporate and assess the utility of admission prediction score in the ED.

## Supporting information

**S1 Table. Details of the area under the curve.**
(DOCX)

## Author Contributions

**Conceptualization:** Muhammad Zahid.

**Data curation:** Prem Chandra.

**Formal analysis:** Prem Chandra, Rajvir Singh.

**Investigation:** Prem Chandra.

**Methodology:** Muhammad Zahid.

**Writing – original draft:** Muhammad Zahid, Adeel Ahmad Khan, Fateen Ata.

**Writing – review & editing:** Muhammad Zahid, Adeel Ahmad Khan, Fateen Ata, Zohaib Yousaf, Vamanjore Aboobacker Naushad, Nishan K. Purayil, Anand Bhaskaran Kartha, Abdelnaser Y. Awad Elzouki, Dabia Hamad S. H. Al Mohanadi, Ahmed Ali A. A. Al-Mohammed.

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
