## [Decision Letter · Decision Letter 0]

2 Aug 2023

PONE-D-23-14864Medical Admission Prediction Score (MAPS) - A simple tool to predict medical admissions.PLOS ONE

Dear Dr. Khan,

Thank you for submitting your manuscript to PLOS ONE. After careful consideration, we feel that it has merit but does not fully meet PLOS ONE’s publication criteria as it currently stands. Therefore, we invite you to submit a revised version of the manuscript that addresses the points raised during the review process.

We look forward to receiving your revised manuscript.

Kind regards,

Reaz Mahmud, MBBS, FCPS (Medicine), MD (Neurology)

Academic Editor

PLOS ONE

Journal Requirements:

"Publication charges were provided by the Qatar National Library (QNL)."

5. We notice that your supplementary table 1 is included in the manuscript file. Please remove them and upload them with the file type 'Supporting Information'. Please ensure that each Supporting Information file has a legend listed in the manuscript after the references list.

**Additional Editor Comments:**

Please add "in the emergency department" at the end of the title. The information presented in lines 77-78 is unnecessary as it is duplicated in lines 79-80. In line 87, the symbol "%" is missing after the sensitivity value. The proper format for in-text citation is [3,44], not as superscript. Instead of "busiest," it is recommended to use "one of the busiest hospitals" in line 132. Please clarify why the age was set to >14, as in most instances, the adult population is defined as >18 years.

After the first citation in the text, please provide the tables. Ensure to follow the journal's requirements for the appropriate font, font size, and spacing in the text, as well as the font size of the headings and subheadings. It is essential to read the journal instructions carefully.

On line 253, the sensitivity value has a typo with the "%" symbol missing. On lines 264-268, it is recommended to avoid duplicating the results in the discussion section and instead provide a summary. Additionally, on line 275, there is a conflict with the term MEWS used.

Please ensure that the conclusion is stated precisely. Please note that volume numbers need to be included for references 6 and 7. When referencing a work with six or more authors, please use "et al." after the sixth author's name. The title of each figure should be positioned below the corresponding figure, and figure legends must be included to explain the content of each figure. Furthermore, figure captions should be inserted in the manuscript's text immediately after the paragraph in which the figure is first mentioned. Lastly, kindly add legends to the table to clarify the different variables.

Reviewers' comments:

Reviewer's Responses to Questions

**Comments to the Author**

1. Is the manuscript technically sound, and do the data support the conclusions?

Reviewer #1: Partly

Reviewer #2: Yes

Reviewer #3: Partly

2. Has the statistical analysis been performed appropriately and rigorously? 

Reviewer #1: Yes

Reviewer #2: Yes

Reviewer #3: Yes

3. Have the authors made all data underlying the findings in their manuscript fully available?

Reviewer #1: No

Reviewer #2: Yes

Reviewer #3: Yes

4. Is the manuscript presented in an intelligible fashion and written in standard English?

Reviewer #1: Yes

Reviewer #2: Yes

Reviewer #3: Yes

5. Review Comments to the Author

Reviewer #1: The authors addressed one of the major problems (overcrowding) in emergency departments by analyzing the data of a large population. The admission prediction model developed in this study seems reasonably helpful in clinical decision making, although physicians should not overly rely on it based on its limited sensitivity and specificity.

Upon reviewing this manuscript, the reviewer found several points which should be considered prior to publication.

First, the authors focused on prediction of hospitalization in the setting of initial ED encounter. However, in ED practice, it may be more meaningful to identify those who are in a potentially life-threatening condition rather than those who need hospitalization. In this view point, it may be worth considering to assess the performance of the scoring system for detection of those who died, required ICU care, or developed complications as well.

Although the admission prediction model should support decision making process, it is not obvious if it really shortens the time until patient admission and solve overcrowding of ED. This should be discussed more.

In real world situations, patients who leave the ED or die in the ED before a decision of admission can be made should also undergo the same clinical evaluation as those who can complete clinical decision making. However, the number of those who had been excluded from the analysis was not shown in the manuscript and it is difficult for readers to judge if this study can be readily generalizable.

In Table 1, “Number of comorbidities” of “Study Cohort” is displayed as “NA.” Are there any reasons for this?

In Table 3, “Regression Coefficient” of reference variables are written as “1.” However, it seems misleading to define the regression coefficients for reference variables as 1 in multivariable logistic regression models. For example, in rows on “Age,” patients “>60 years” have higher odds of admission with OR 1.88 (1.77-2.01) as compared to “≦60 years.” But regression coefficient of “≦60 years” is 1.0 and higher than 0.629 for “>60 years.” Such expressions seem confusing.

The methods of determining the best-suitable cutoff of the scoring system should be elaborated more (such as Youden index or closest to (0, 1) approach etc).

In terms of data availability, the original data or their location could not be found in the manuscript. Instead, “Data Sharing” section says “The datasets used and/or analysed in the current study are available from the corresponding author on reasonable request conditional to the permission from the Medical Research Centre.” If so, it seems more appropriate to answer “No” to “Data Availability” question (because there seems some restriction) and elaborate the reason as needed.

Minor issue:

In line 246, the expression “independently associated” should not be used while talking about univariate analysis.

Reviewer #2: I congratulate researchers for conducting research with large amounts of data. There are some minor additions:

1. In the discussion of line 266, it is better to add the criteria for AUC which is said to have good accuracy (eg AUC > 0.9 has high accuracy) and its source.

2. In conclusion, it should be added that this score can be used in patients aged > 14 years because your research inclusion is > 14 years old

Reviewer #3: The manuscript seeks to provide and easy to used scoring system to guide admission decision making at emergency departments. The result shows a good potential of the developed scoring system. However, it is a little farfetched to conclude that the results have better generalizability considering that about 92% of the population are Asians and Arabs. Below are a few other issues that needs to be addressed to improve the quality of the manuscript.

1. Although the term “Medical Admission Prediction Score (MAPS)” is used in the title, it is not used in the content of the manuscript. In lines 279-289, you rather referred to your model as MEWS

2. In the abstract section, the total number of presentations in the methods and in the results do not match

3. In lines 111 to 112 the authors should clarify if the setting of the report is the USA or global

4. The inclusion and exclusion criteria could be added to the Study population section since they were introduced there

5. The sentence "We abstracted clinical, demographic and laboratory data of the patients from the EMR and the patients were not contacted directly to provide any information" could be modified as "Extracted data from the medical records include clinical, Demographic and laboratory data"

6. The authors should provide some justification on why patients 14 years and below were excluded

7. Lines 169 -171 is already captured in lines 133-137. It therefore not necessary to repeat it

8. How were missing data handled in the analysis?

9. The authors should check line 177 -180 and correct the types of data and variables and the required test used for the analysis. For instance, it states that associations between qualitative data was assessed using chi-square or Fisher's exact test. This is not accurate.

10. The authors should define the groups that are being compared and also indicate the list of variables that were included in the regression models in the statistical analysis section

11. Lines 181 - 190 should be moved to last section of the paragraph for coherence

12. Only one figure is presented so the sentence "Pictorial presentations of the key results were made using appropriate statistical graphs" may not be necessary

13. The sentence in lines 206 -208 need to be restructured appropriately for clarity

14. Avoid beginning sentences with figures. Also be consistent on how the figures are written, some have commas others do not.

15. Delete "inpatient" in line 261

16. Provide some background information on what MEWS is used for, for context in the discussion

17. Delete "to calculate MEWS" in lines 280 as it introduces some confusion as to whether the authors are revising MEWS or comparing their model to MEWS. Otherwise the authors should provide more clarity

18. In the discussion section too much attention is paid to the history of admission within the last 30 days. The other variables in the model were not highlighted

19. The authors need to pay close attention to the mathematical symbols used in the data categorisation as indicated in the tables. Its current categorisation leaves out some values. For instance, age was categorised as <60 years and >60 years. This leaves out all those who are exactly 60 years old. Similar errors are found in the other categories and may have impacted the data analysis and findings. Also, the appropriate sign less than or equal to should be used thus "≥" instead of ">= "

20. Each table should have the definition of all abbreviations used underneath it. For each variable, the authors should also define the test used to compare the groups

6. PLOS authors have the option to publish the peer review history of their article (what does this mean?). If published, this will include your full peer review and any attached files.

Reviewer #1: No

Reviewer #2: No

Reviewer #3: No

---

## [Author Response · Author response to Decision Letter 0]

29 Aug 2023

Dear Editor and reviewers,

Thank you for taking the time to review our manuscript. Your comments have significantly contributed to improving the quality of this manuscript. Following is the point-by-point response of the author panel to the reviewer’s comments. 

In case of any further queries, please do not hesitate to contact us.

Dr. Adeel Ahmad Khan

Author’s Response: Edited as advised.

"Publication charges were provided by the Qatar National Library (QNL)"

Author’s Response: The funding statement has been updated as advised.

Author’s Response: Edited as advised.

Author’s Response: Edited as advised. 

5. We notice that your supplementary table 1 is included in the manuscript file. Please remove them and upload them with the file type 'Supporting Information'. Please ensure that each Supporting Information file has a legend listed in the manuscript after the references list.

Author’s Response: Edited as advised.

Author’s Response: Edited as advised.

Additional Editor Comments:

Please add "in the emergency department" at the end of the title. The information presented in lines 77-78 is unnecessary as it is duplicated in lines 79-80. In line 87, the symbol "%" is missing after the sensitivity value. The proper format for in-text citation is [3,44], not as superscript. Instead of "busiest," it is recommended to use "one of the busiest hospitals" in line 132. Please clarify why the age was set to >14, as in most instances, the adult population is defined as >18 years.

Author’s Response: Edited as advised. 

In Qatar, at Hamad Medical Corporation, the adult population is considered above 14 years of age. That is why we included all patients aged >14 years.

After the first citation in the text, please provide the tables. 

Author’s Response: We are not able to understand this comment by the reviewer. Please elaborate. Thank you

Ensure to follow the journal's requirements for the appropriate font, font size, and spacing in the text, as well as the font size of the headings and subheadings. It is essential to read the journal instructions carefully.

Author’s Response: The journal instructions state that “Use a standard font size and any standard font, except for the font named Symbol.” The font size used is size 11 in our manuscript. 

On line 253, the sensitivity value has a typo with the "%" symbol missing. On lines 264-268, it is recommended to avoid duplicating the results in the discussion section and instead provide a summary. Additionally, on line 275, there is a conflict with the term MEWS used.

Author’s Response: Edited as advised.

Regarding MEWS, there has been confusion due to the way it has been written by us. MEWS is not our score. MEWS is another score that has been published in the literature and consists of vital signs and is used to predict admission. MEWS uses only vital signs in predicting admission. We have also used the vital signs (similar to MEWS) in addition to several other variables to calculate our medical admission prediction score (MAPS). We have edited the paragraph for more clarity.

Please ensure that the conclusion is stated precisely. Please note that volume numbers need to be included for references 6 and 7. When referencing a work with six or more authors, please use "et al." after the sixth author's name. The title of each figure should be positioned below the corresponding figure, and figure legends must be included to explain the content of each figure. Furthermore, figure captions should be inserted in the manuscript's text immediately after the paragraph in which the figure is first mentioned. Lastly, kindly add legends to the table to clarify the different variables.

Author’s Response: Edited as advised

Reference 6 and 7 are from the web pages, and hence volume numbers are not available. The webpage link has been provided.

Reviewer #1: The authors addressed one of the major problems (overcrowding) in emergency departments by analyzing the data of a large population. The admission prediction model developed in this study seems reasonably helpful in clinical decision making, although physicians should not overly rely on it based on its limited sensitivity and specificity.

Upon reviewing this manuscript, the reviewer found several points which should be considered prior to publication.

First, the authors focused on prediction of hospitalization in the setting of initial ED encounter. However, in ED practice, it may be more meaningful to identify those who are in a potentially life-threatening condition rather than those who need hospitalization. In this view point, it may be worth considering to assess the performance of the scoring system for detection of those who died, required ICU care, or developed complications as well.

Author’s response: Thank you for providing a valuable suggestion. We agree with the reviewer that recognizing patients with life-threatening is a very important aspect that needs to be addressed. However, the current paper focusses on a model that predicts medical admissions from the ED and hence, assessment of scoring system to detect life-threatening condition, mortality, ICU admission and complcations is out of the scope of this paper.

Although the admission prediction model should support decision making process, it is not obvious if it really shortens the time until patient admission and solve overcrowding of ED. This should be discussed more.

In real world situations, patients who leave the ED or die in the ED before a decision of admission can be made should also undergo the same clinical evaluation as those who can complete clinical decision making. However, the number of those who had been excluded from the analysis was not shown in the manuscript and it is difficult for readers to judge if this study can be readily generalizable.

Author’s response: 

We acknowledge that the effect of admission prediction score on ED duration of stay and overcrowding cannot be assessed from this study due to its retrospective design. We have made this further clear in the limitations section.

A flowsheet showing the process of inclusion of patients and reasons for exclusion has been added as Figure 1.

In Table 1, “Number of comorbidities” of “Study Cohort” is displayed as “NA.” Are there any reasons for this?

Authors’ Response: Number of comorbidities of the whole cohort have been added

In Table 3, “Regression Coefficient” of reference variables are written as “1.” However, it seems misleading to define the regression coefficients for reference variables as 1 in multivariable logistic regression models. For example, in rows on “Age,” patients “>60 years” have higher odds of admission with OR 1.88 (1.77-2.01) as compared to “≦60 years.” But regression coefficient of “≦60 years” is 1.0 and higher than 0.629 for “>60 years.” Such expressions seem confusing.

The methods of determining the best-suitable cutoff of the scoring system should be elaborated more (such as Youden index or closest to (0, 1) approach etc).

Authors’ Response: Kindly note that the value presented as ‘1’ mainly to indicated reference category rather to be taken as respective regression coefficients values. Thus, to make this more precise, we have removed value ‘1’ from this table 3 and it is now being presented as “Reference category”. 

In terms of data availability, the original data or their location could not be found in the manuscript. Instead, “Data Sharing” section says “The datasets used and/or analysed in the current study are available from the corresponding author on reasonable request conditional to the permission from the Medical Research Centre.” If so, it seems more appropriate to answer “No” to “Data Availability” question (because there seems some restriction) and elaborate the reason as needed.

Author’s Response: Data availability statement has been updated

Minor issue:

In line 246, the expression “independently associated” should not be used while talking about univariate analysis.

Author’s Response: Edited as advised.

Reviewer #2: I congratulate researchers for conducting research with large amounts of data. There are some minor additions:

1. In the discussion of line 266, it is better to add the criteria for AUC which is said to have good accuracy (eg AUC > 0.9 has high accuracy) and its source.

Author's Response: 

The following refence for AUC classification/criteria has now been added in the manuscript.

Polo TCF, Miot HA. Use of ROC curves in clinical and experimental studies. J Vasc Bras. 2020;19: e20200186. https://doi.org/10.1590/1677-5449.200186 (correct the references)

2. In conclusion, it should be added that this score can be used in patients aged > 14 years because your research inclusion is > 14 years old

Author’s Response: Edited as advised

Reviewer #3: The manuscript seeks to provide and easy to used scoring system to guide admission decision making at emergency departments. The result shows a good potential of the developed scoring system. However, it is a little farfetched to conclude that the results have better generalizability considering that about 92% of the population are Asians and Arabs. 

Author’s Response: This is a general comment by the reviewer and does not require any specific response.

Below are a few other issues that needs to be addressed to improve the quality of the manuscript.

1. Although the term “Medical Admission Prediction Score (MAPS)” is used in the title, it is not used in the content of the manuscript. In lines 279-289, you rather referred to your model as MEWS

Author’s Response: Thank you for your comment. We believe there has been confusion due to the way it has been written by us. MEWS is not our score. MEWS is another score that has been published in literature and consists of vital signs and is used to predict admission. MEWS uses only vital signs in predicting admission. We have also used the vital signs (similar to MEWS) in addition to several other variables to calculate our medical admission prediction score (MAPS). We have edited the paragraph for more clarity

2. In the abstract section, the total number of presentations in the methods and in the results do not match

Author’s Response: Thank you for pointing out the mistake. The number in results section is correct. The line from methods section has been removed anyways on advise of another reviewer.

3. In lines 111 to 112 the authors should clarify if the setting of the report is the USA or global

Author’s Response: Edited as advised

4. The inclusion and exclusion criteria could be added to the Study population section since they were introduced there

Author’s Response: Edited as advised

5. The sentence "We abstracted clinical, demographic and laboratory data of the patients from the EMR and the patients were not contacted directly to provide any information" could be modified as "Extracted data from the medical records include clinical, Demographic and laboratory data"

Author’s Response: The line has been deleted as per your suggestion in a later comment.

6. The authors should provide some justification on why patients 14 years and below were excluded

Author’s Response: In Hamad Medical Corporation, adult population is defined as age ≥ 14 years. Hence, patients less than 14 years age were excluded. In the study population section, we have mentioned clearly that “Pre-specified data of all adult patients who attended…..”

7. Lines 169 -171 is already captured in lines 133-137. It therefore, not necessary to repeat it

Author’s Response: Deleted as advised

8. How were missing data handled in the analysis?

Author’s Response: Thank you. Kindly note that the demographic and comorbidities related information were available in all the cases. However, information on vitals, GCS and discharge within 30 days noted to be missing in <4% of the cases except GCS in which 6.5% cases had missing information. We strongly believe that with such extremely low missing observations particularly with current such larger database/observations would not have had any potential impact on either statistical significance derived or inferential analysis performed. 

9. The authors should check line 177 -180 and correct the types of data and variables and the required test used for the analysis. For instance, it states that associations between qualitative data was assessed using chi-square or Fisher's exact test. This is not accurate.

Author’s Response: Thank you for pointing out this mistake. We have rechecked the statistical analysis section and edited where needed. Thank you.

10. The authors should define the groups that are being compared and also indicate the list of variables that were included in the regression models in the statistical analysis section

Author’s Response: Done. Thank you.

11. Lines 181 - 190 should be moved to last section of the paragraph for coherence

Author’s Response: Edited as advised

12. Only one figure is presented so the sentence "Pictorial presentations of the key results were made using appropriate statistical graphs" may not be necessary

Author’s Response: Deleted as advised

13. The sentence in lines 206 -208 need to be restructured appropriately for clarity

Author’s Response: Edited as advised

14. Avoid beginning sentences with figures. Also be consistent on how the figures are written, some have commas others do not.

15. Delete "inpatient" in line 261

Author’s Response: Deleted as advised

16. Provide some background information on what MEWS is used for, for context in the discussion

Author’s Response: Edited as advised

17. Delete "to calculate MEWS" in lines 280 as it introduces some confusion as to whether the authors are revising MEWS or comparing their model to MEWS. Otherwise the authors should provide more clarity

Author’s Response: Edited as advised

18. In the discussion section too much attention is paid to the history of admission within the last 30 days. The other variables in the model were not highlighted

Author’s Response: This variable has not been part of most of the other scoring systems in the literature. Hence, we have focused a lot on it during the discussion as it adds value to our scoring system

19. The authors need to pay close attention to the mathematical symbols used in the data categorisation as indicated in the tables. Its current categorisation leaves out some values. For instance, age was categorised as <60 years and >60 years. This leaves out all those who are exactly 60 years old. Similar errors are found in the other categories and may have impacted the data analysis and findings. Also, the appropriate sign less than or equal to should be used thus "≥" instead of ">= "

Author’s Response: Thank you for pointing that out. Symbols have been corrected

20. Each table should have the definition of all abbreviations used underneath it. For each variable, the authors should also define the test used to compare the groups

Author’s Response: Done. Thank you.

---

## [Decision Letter · Decision Letter 1]

6 Oct 2023

Medical Admission Prediction Score (MAPS); A simple tool to predict medical admissions in the emergency department.

PONE-D-23-14864R1

Dear Dr. Khan,

We’re pleased to inform you that your manuscript has been judged scientifically suitable for publication and will be formally accepted for publication once it meets all outstanding technical requirements.

Kind regards,

Reaz Mahmud, MBBS, FCPS (Medicine), MD (Neurology)

Academic Editor

PLOS ONE

Additional Editor Comments (optional):

Reviewers' comments:

Reviewer's Responses to Questions

**Comments to the Author**

1. If the authors have adequately addressed your comments raised in a previous round of review and you feel that this manuscript is now acceptable for publication, you may indicate that here to bypass the “Comments to the Author” section, enter your conflict of interest statement in the “Confidential to Editor” section, and submit your "Accept" recommendation.

Reviewer #1: All comments have been addressed

Reviewer #4: All comments have been addressed

2. Is the manuscript technically sound, and do the data support the conclusions?

Reviewer #1: Yes

Reviewer #4: Yes

3. Has the statistical analysis been performed appropriately and rigorously? 

Reviewer #1: Yes

Reviewer #4: Yes

4. Have the authors made all data underlying the findings in their manuscript fully available?

Reviewer #1: Yes

Reviewer #4: Yes

5. Is the manuscript presented in an intelligible fashion and written in standard English?

Reviewer #1: Yes

Reviewer #4: Yes

6. Review Comments to the Author

Reviewer #1: The authors addressed all the previously raised comments appropriately. The manuscript now seems acceptable for publication.

Reviewer #4: The authors have diligently addressed all reviewer comments, significantly enhancing the study's quality. The well-executed research, sound analysis, and clear writing make it suitable for publication.

7. PLOS authors have the option to publish the peer review history of their article (what does this mean?). If published, this will include your full peer review and any attached files.

Reviewer #1: No

Reviewer #4: No

---

## [Editor Report · Acceptance letter]

16 Oct 2023

PONE-D-23-14864R1 

Medical Admission Prediction Score (MAPS); A simple tool to predict medical admissions in the emergency department. 

Dear Dr. Khan:

I'm pleased to inform you that your manuscript has been deemed suitable for publication in PLOS ONE. Congratulations! Your manuscript is now with our production department. 

Kind regards, 

on behalf of

Dr. Reaz Mahmud 

Academic Editor

PLOS ONE